# Functionalization of Polyethylene (PE) and Polypropylene (PP) Material Using Chitosan Nanoparticles with Incorporated Resveratrol as Potential Active Packaging

**DOI:** 10.3390/ma12132118

**Published:** 2019-07-01

**Authors:** Tjaša Kraševac Glaser, Olivija Plohl, Alenka Vesel, Urban Ajdnik, Nataša Poklar Ulrih, Maša Knez Hrnčič, Urban Bren, Lidija Fras Zemljič

**Affiliations:** 1Laboratory for Characterization and Processing of Polymers, Faculty of Mechanical Engineering, University of Maribor, Smetanova 17, SI-2000 Maribor, Slovenia; 2Department of Surface Engineering and Optoelectronics, Jožef Stefan Institute, Teslova 30, SI-1000 Ljubljana, Slovenia; 3Department of Food Science and Technology, Biotechnical Faculty, University of Ljubljana, Jamnikarjeva 101, SI-1000 Ljubljana, Slovenia; 4Faculty of Chemistry and Chemical Engineering, University of Maribor, Smetanova 17, SI-2000 Maribor, Slovenia

**Keywords:** nanoparticles, chitosan-resveratrol, PE/PP functionalization, O_2_ plasma, active packaging

## Abstract

The present paper reports a novel method to improve the properties of polyethylene (PE) and polypropylene (PP) polymer foils suitable for applications in food packaging. It relates to the adsorption of chitosan-colloidal systems onto untreated and oxygen plasma-treated foil surfaces. It is hypothesized that the first coated layer of chitosan macromolecular solution enables excellent antibacterial properties, while the second (uppermost) layer contains a network of polyphenol resveratrol, embedded into chitosan nanoparticles, which enables antioxidant and antimicrobial properties simultaneously. X-ray photon spectroscopy (XPS) and infrared spectroscopy (FTIR) showed successful binding of both coatings onto foils as confirmed by gravimetric method. In addition, both attached layers (chitosan macromolecular solution and dispersion of chitosan nanoparticles with incorporated resveratrol) onto foils reduced oxygen permeability and wetting contact angle of foils; the latter indicates good anti-fog foil properties. Reduction of both oxygen permeability and wetting contact angle is more pronounced when foils are previously activated by O_2_ plasma. Moreover, oxygen plasma treatment improves stability and adhesion of chitosan structured adsorbates onto PP and PE foils. Foils also exhibit over 90% reduction of *Staphylococcus aureus* and over 77% reduction of *Escherichia coli* as compared to untreated foils and increase antioxidant activity for over a factor of 10. The present method may be useful in different packaging applications such as food (meat, vegetables, dairy, and bakery products) and pharmaceutical packaging, where such properties of foils are desired.

## 1. Introduction

Recently, novel-packaging materials have become an essential part of the global food packaging market aiding the ever-increasing consumer awareness and importance of eco-friendly substitutes. The demand for innovative packaging is expanding and will continue to grow as the companies utilize packaging as a medium to protect their products and to promote environment protection [1,2,3,4,5]. The European packaging industry has a market value of approximately 80 billion EUR and accounts for ≈40% of the global packaging market. The future trend is oriented toward growing markets on a global scale [2]. Major manufacturers of packaging materials are looking to differentiate their products from those of their competitors by providing the best possible functional and/or biodegradable packaging products as per consumer demands. To be competitive, it is important that new packaging solutions are bio-based, multifunctional, economically viable, and easily incorporated into established industrial manufacturing processes. Sufficient competitiveness can only be realized if the EU food industry speeds up the pace of innovation [1,2,3,4,5]. Over recent decades, majority of research on the development of innovative food packaging materials has been carried out with a view to combat against pathogens, to reduce spoilage and waste, to optimize process efficiency, to reduce the need for chemical preservatives, to improve the functionality of foods, and to improve the nutritional and sensorial properties of food, responding to the demands of different consumer niches and markets, also in terms of affordability [1,2,3,4,5,6,7].

However, few of these investigations were economical, eco-friendly, and healthy; thus, it is still a great challenge to be involved in the study of a new concept of active packaging material [1,2,3,4,5,6,7]. Beside increasing public health awareness, the changes in retailing practices (such as market globalization resulting in longer distribution of food) and fact that the industry must follow the strict EU packaging guidelines and regulations, all act as a driving force for a significant interest to incorporate natural antimicrobial ingredients into a basic packaging material and, in this way, developing an active concept of packaging [1,2,3,4,5,6,7].

Examples of prospective natural biodegradable and bioactive substances are less-employed antimicrobial polysaccharides and their derivatives. Chitosan, obtained from chitin by partial deacetylation, which makes it soluble in acidic aqueous solutions, has diverse, industrially important properties, such as biocompatibility, biodegradability, and non-toxicity. It is widely known as a valuable material for different biomedical and food applications [3,7,8,9,10,11,12]. There are numerous studies on utilizing chitosan as coating for potential packaging material [13,14,15,16,17,18,19,20,21]. However, in spite of chitosan’s excellent antimicrobial activity, it has been found that chitosan shows very poor antioxidant activity [22,23]. The latter is required for a bioactive packaging material in order to impede the natural processes, as food spoilage (i.e., the oxidative deterioration of meat products is caused by the degradation reactions of fats and pigments) is diminished by reduction of oxygen and moisture. Thus, it is greatly appreciated to increase antioxidant activity of chitosan with the natural compounds including proven high antioxidant capacity, such as plant phenolics. Tocopherols are the only natural food grade antioxidants regulated by the Food and Drug Administration (FDA). Spices and flavorings like rosemary and sage are regulated only as flavorings. An ever-popular trend is the use of vegetable and essential oils as natural antioxidants [24,25,26]. Some plant extracts such as grapefruit seed, cinnamon, horseradish, clove, garlic, and many others have been previously added to a packaging system to demonstrate effective antioxidant–antimicrobial activity against spoilage and pathogenic bacteria [27,28,29,30,31,32]. Today, resveratrol (RES) is believed to be one of the most potent polyphenols and strongest protectors against symptoms, associated with aging and free radical damage [33,34]. Resveratrol has already been combined with chitosan with the aim to improve the solubility, stability, and cellular uptake of RES by nanoencapsulation using chitosan (CS) and γ-poly (glutamic acid) (γ-PGA) [35]. Another study suggests that chitosan–sodium tripolyphosphate (TPP) microspheres could be used as a potential delivery system with controlled release of resveratrol [36]. Active linear low-density polyethylene (LLDPE) composites were manufactured by immobilization of RES in the polymeric matrix and by pre-incorporation of the RES into a food-contact-permitted montmorillonite clay prior to melt mixing with the polymer. The resulting composites not only showed strong antioxidant activity, but also antimicrobial activity. In addition, the thiobarbituric acid reactive substances method (TBARS) was applied to assess the comparative oxidative behavior of fresh meat in air and direct contact with the LLDPE film containing the active clay. The results suggest that this technology could potentially extend the shelf-life of red meat over a few days by a mechanism of free radicals trapping and the subsequent disabling of food oxidation processes [37]. Active, cellulose derivative-based films (hydroxyethyl cellulose and cellulose acetate) were developed and prepared by the casting method using RES and its inclusion into a complex with hydroxypropyl-γ-cyclodextrin as an active agent was demonstrated. The structural, mechanical, barrier, surface free energy, optical, and release properties were analyzed. The antimicrobial activity of these films against *Campylobacter jejuni, Campylobacter coli,* and *Arcobacter butzleri* were evaluated by the agar diffusion method. The anti-quorum sensing (QS) activity of films was studied using biosensor strain *Chromobacterium violaceum* ATCC 12472. The active films developed in this work were transparent, inhibited the growth of the tested strains, and possessed anti-QS activity as well. This study showed the potential of these active films as a new approach to control the growth of *Campylobacter genus* [38].

However, to the best of our knowledge, until now, there was no methodologic functionalization and detailed characterization of PE and PP foils, as the most representative packaging material, functionalized by chitosan with incorporated resveratrol, showed clear and highly significant advances compared to the other relevant studies. Because PE and PP materials are inert and hydrophobic, the adsorption of chitosan-colloidal systems onto its surfaces may be improved using different activation procedures in order to obtain carboxylic groups on its surface, that represent binding places for chitosan amino groups (Coulomb interactions). Plasma activation can tune hydrophobic character into hydrophilic and, thus, increase the ability of PE and PP for improved attachment of chitosan. In our previous work, we have already shown and discussed that advanced and environmentally friendly cold oxygen plasma treatment can be used to enhance the adhesion of chitosan [16]. This paper introduces the development of complex, composite material based on previous PE/PP-cold oxygen plasma activation and further attachment of chitosan colloidal systems in different structural forms; i.e., (i) chitosan macromolecular solution was applied as a first layer onto PE and PP foils and, (ii) RES was encapsulated into chitosan nanoparticles and further applied onto i) coated PE and PP foils as an upper layer (layer-by-layer deposition).

The functionalized foils were physicochemical analyzed from the following point of view: (i) Gravimetric, (ii) surface elemental composition of functionalized foils, (iii) oxygen permeability, (iv) wettability, and (v) morphology. In order to obtain migration profile of chitosan and resveratrol from foils surface, desorption studies were done using polyelectrolyte titration. Most importantly, bioactive properties of foils were also examined. The antimicrobial properties were analyzed using the standard ISO 22196, whereas the anti-oxidative properties were determined spectrophotometrically using ABTS test.

## 2. Materials and Methods

### 2.1. Materials

Low-molecular weight chitosan (50 to 190 kDa), poly (D-glucosamine) from Sigma-Aldrich; sodium tripolyphosphate—TPP (MW = 367.86 g/mol) from Sigma-Aldrich, St. Louis, MI, USA; Resveratrol (MW = 228.24 g/mol), ≥99% (HPLC) from Sigma-Aldrich; acetic acid (MW = 60.05 g/mol), ≥99.8% from Sigma-Aldrich; ethanol (99.8%, GC) from Sigma-Aldrich. Ultrapure water—Milli-Q Direct system (Millipore, Burlington, MA, USA). Polyethylene—PE normal quality, transparent, GSM = 46.28 g/m² (thickness 50 µm, Slippery 0.207) from Makoter d.o.o., Ljutomer, Slovenia; Polypropylene—PP normal transparent oriented, GSM = 22.93 g/m² (Thickness 27 µm, Slippery 0.278) from Manucor S.p.A., Sessa Aurunca, Italy.

### 2.2. Solutions/Dispersions

#### 2.2.1. Preparation of Chitosan Solutions

Different concentrations of chitosan solutions (1% and 2%, w/v), were prepared by adding chitosan powder in ultrapure water under constant magnetic stirring, while acetic acid was added drop-wise to enable dissolution of chitosan. The solution was left stirring overnight and pH was adjusted with acetic acid to 4.0.

#### 2.2.2. Preparation of TPP Solution

Sodium tripolyphosphate (TPP) powder was suspended in ultrapure water in order to prepare a 0.2% (w/v) solution.

#### 2.2.3. Preparation of Resveratrol Solution

Resveratrol (RES) powder was suspended in absolute ethanol in order to prepare a 12.5 mg/mL solution.

#### 2.2.4. Preparation of Resveratrol-Loaded Chitosan-TPP Nano Dispersion (i.e., Chitosan–Resveratrol Nano Dispersion—CSNPs RES)

Chitosan nanoparticles were prepared by the ionic gelation technique. Simultaneously, 0.2% (w/v) of TPP solution and 12.5 mg/mL resveratrol solutions were added to a fixed volume of 1% (w/v) chitosan solution in order to obtain 5:1 chitosan to TPP weight ratio. This ratio was chosen according to the previously published work, reporting it as an optimal ratio to obtain desirable antimicrobial activity of nanoparticles’ dispersion [39]. Particles were formed spontaneously under constant stirring for 1 h at room temperature. The final pH of CS nanoparticles dispersions with RES (CSNPs RES) was adjusted to 4.0 with acetic acid.

### 2.3. Functionalization of Foils

#### 2.3.1. Pre-Treatment of PE and PP with Plasma

PE and PP foils were cleaned, dried, and cut to the size of the print. Foils were activated with oxygen microwave plasma in a surfatron mode. The forward power was set to 200 W. Oxygen pressure in the treatment chamber made of quartz glass was set to 50 Pa. The samples were exposed to late oxygen afterglow for 60 s.

#### 2.3.2. Functionalization of PE and PP Surface with Chitosan, Chitosan–Resveratrol Nano Dispersion (CSNPs RES dispersion)

For antimicrobial and anti-oxidative functionalization of PE and PP surface (untreated and oxygen plasma treated), 2% chitosan and CSNPs RES dispersions were used. The coating was applied in two layers (layer-by-layer deposition). The first layer was made of 2% chitosan, which serves for better adhesion and a higher antimicrobial effect. The second (upper) layer was formed with CSNPs RES dispersion. PE and PP foils were previously cleaned, dried, cut to the size of the print, and weighed. For the application of functional coatings, the method of printing through the fine fabric using a magnet was applied (roll printing). The foils were dried at each step at room temperature. Sample descriptions and notations are given in the Table 1.

### 2.4. Methods

#### 2.4.1. Dispersions Characterization

##### Particle Size, PDI, and Zeta Potential Determination

The particles hydrodynamic diameter and polydispersity index (PDI) of prepared nanoparticles dispersion were determined by dynamic light scattering—DLS (Zetasizer Nano ZS, Malvern Instruments Ltd., Malvern, UK)—at a temperature of 25 °C. Zeta potential (ZP) was determined by performing an electrophoresis experiments and measuring the velocity of the particles using laser doppler velocimetry (LDV) on Zetasizer Nano ZS (Malvern Instruments Ltd.). Before analyses, dispersion was stirred for 15 min and adjusted to pH 4 with acetic acid if it was necessary. For carrying out the DLS measurements, the disposable cuvettes were used and for ZP determination disposable folded capillary cell with electrodes. Data were collected using Malvern Zetasizer Software version 7.12.

##### Minimal Inhibitory Concentration (MIC)

MIC of macromolecular chitosan solution, nanoparticles dispersion, and resveratrol solution was defined as the lowest concentration that allowed no more than 20% growth of microbes. Three methods (disk diffusion method, agar dilution method, broth microdilution method) were used for MIC determination, while the most useful was the microdilution method. Therefore, it was chosen as protocol for our samples.

For the test, 50 μL of each bacterial suspension in a suitable growth medium was added to the wells of a sterile 96-well microtiter plate already containing 50 μL of two-fold serially diluted chosen substrate in a proper growth medium. The final volume was 100 μL. Control wells were prepared with a culture medium, bacterial suspension only, chitosan solution, chitosan nanoparticles dispersion, and resveratrol solution only and ethanol in amounts corresponding to the highest quantity present in resveratrol solution. The contents of each, respectively, were well mixed on a microplate shaker at 900 rpm for 1 min prior to incubation for 24 h in the cultivation conditions described above [40]. The MIC was the lowest concentration where no viability was observed after 24 h because of metabolic activity [41]. To indicate respiratory activity, the presence of color was determined after adding 10 μL/well of INT (2-p-iodophenyl-3-pnitrophenyl-5-phenyl tetrazolium chloride, Sigma) or TTC (2,3,5-triphenyl tetrazolium chloride) dissolved in water (INT 2 mg/mL, TTC 20 mg/mL) and incubated under appropriate cultivation conditions for 30 min in the dark [42]. To determine the ATP activity, the bioluminescence signal was measured by a Microplate Reader after adding 100 μL/well of BacTiter-Glo™ reagent and after 5 min incubation in the dark. Positive controls were wells with a bacterial suspension in an appropriate growth medium and a bacterial suspension in an appropriate growth medium with ethanol in amounts corresponding to the highest quantity present in the broth microdilution assay. Negative controls were wells with growth medium and chitosan substrates and resveratrol. All measurements of MIC values were repeated in triplicate [40].

#### 2.4.2. Functionalized Foils

##### Gravimetric Measurements of Sample Mass

All samples were weighed (four decimal places) to compare their weights with untreated reference foils, which were previously cleaned, dried, and cut to the size of the print. The final weight differences between the reference foils and applied foils showed a successful application of the formulation onto the PE and PP foils. The final weight differences between the reference foils and functionalized foils were calculated for absolute dry samples.

##### Surface Elemental Composition of Functionalized Foils: ATR-FTIR Spectroscopy

The ATR-FTIR spectra were recorded on a Perkin Elmer Spectrum GX NIR FT-Raman spectrometer (Waltham, MA, USA). The ATR accessory contained a diamond crystal. All spectra (16 scans at 4 cm^−1^ resolution, background, and the sample spectra were obtained in the 400–4000 cm^−1^ wavenumber range) were recorded at room temperature. Spectra of samples were deconvoluted with smoothing filter and baseline corrected.

##### Surface Elemental Composition of Functionalized Foils: XPS Analysis

Spectra were recorded using XPS instrument PHI TFA XPS Physical Electronics, Chanhassen, MN, USA in order to assess the surface of the sample. The base pressure in the XPS analysis chamber was approximately 6 × 10^−8^ Pa. The samples were excited with X-rays with monochromatic Al Kα1.2 radiation (1486.6 eV) operating at 200 W. Photoelectrons were detected with a hemispherical analyzer, positioned at an angle of 45° with respect to the normal to the sample surface. The energy resolution was about 0.6 eV. Spectra were recorded from at least two locations on each sample, using analysis area of 400 μm. Surface elemental concentrations were calculated from the survey-scan spectra using the Multipak software version 9.6.0.

##### Oxygen Permeability

The oxygen permeability was determined using Oxygen Transmission Rate System PERME^®^ OX2/230 (Labthink Instruments Co., Ltd., Jinan, China), by standard ASTM D3985. Oxygen transmission rate (OTR) values and coefficient values are averaged results, obtained by two testing of five measurements. All specimens were conditioned at 23 °C and 50% relative humidity 24 h prior testing (flux = 10 mL/min). Thickness of PE and PP foils were measured with calliper at five different places.

##### Goniometry

Contact angles were measured to estimate the surface wettability of coated foils via goniometer DataPhysics, Germany. Static contact angle (SCA) was measured using a drop of liquid resting on the surface. Goniometer with SCA 20 software was used to determine CSA at room temperature with ultrapure water. A small drop (3 µL) of water was carefully placed on the surface of the sample. For each sample, three repetitions were made.

##### Morphology: Scanning Electron Microscopy

For scanning electron microscopy (SEM) imaging of surface morphology, samples of PE and PP foils were prepared by cutting the foils into small, approximately 0.5 × 0.5 cm^2^ square pieces. These pieces were attached to the aluminum sample holders with an adhesive carbon tape to ensure conductivity. For this purpose, Carl Zeiss Supra 35VP scanning electron microscope (Jena, Germany) was employed, where the reference foils and functionalized foils were analyzed at an accelerating voltage of 1 kV and variable working distance using a 10 µm-sized aperture.

##### Migration: Desorption Experiment—Polyelectrolyte Titration

Preparation of desorption bath and samples—0.5 g of PE and PP foils were treated for 24 h with 30 mL of ultrapure water (pH = 5.8, adjusted by 0.1 M hydrochloric acid). The solution was filtered and further used for polyelectrolyte titration to analyze the amount of desorbed chitosan from foils.

For incremental addition of the polyelectrolyte titrant (PES-Na; c = 10 mM) into 30 mL of desorption bath, Mettler Toledo DL 53 titration unit with a 10 mL burette was used. The incremental additions of 0.05 mL were added every 5–10 s. The absorbance was measured as a potential change in mV, using Mettler Toledo DP5 Phototrode at a wavelength of 660 nm. Concentration of the protonated amino groups was determined from the equivalent point of titration curve (Figure 7), as the steepest bit of the curve, and by estimating a 1.1 binding stoichiometry of polyethylenesufonate to chitosan amine groups.

##### Bioactivity

(1) Antimicrobial Activity Testing

For antimicrobial test of functionalized PE and PP foils surface, a modified version of ISO 22196 (plastics—measurement of antibacterial activity on plastics surfaces) was followed [43]. This is currently the test protocol of choice for testing surfaces for antimicrobial effectiveness. Antibacterial activity is described as a state, where growth of bacteria on the surfaces of products is suppressed or as the effect of an agent, which suppresses the growth of bacteria on the surfaces of products. Gram-positive bacteria (*Staphylococcus aureus*) and Gram-negative bacteria (*Escherichia coli*) from the culture collection of Laboratory for Food Microbiology at Department of Food Science and Technology, Biotechnical Faculty, University of Ljubljana were tested. After determination of viable cells number, antimicrobial activity was calculated using Equation (1) and percentage of reduction using Equation (2).
R = Ut − At,(1)
Reduction (%) = ((Ut − At)/Ut) × 100,(2)
where R is the antibacterial activity; Ut is the number of viable cells recovered from the untreated material after incubation; and At is the number of viable cells recovered from the treated material after incubation.

(2) Anti-Oxidative Activity (ABTS)

Anti-oxidative activity was determined using a biochemical reagent, ABTS (2,2′-azino-bis(3-ethylbenzothiazoline-6-sulphonic acid)). The method is based on the reduction of the ABTS•+ radical, which is determined spectrophotometrically at a wavelength of 734 nm. The ABTS•+ cation radical was produced by the reaction between 7 mM ABTS in ultrapure and 2.45 mM potassium persulfate and stored in the dark at room temperature for 12 h. Then, 3.9 mL of diluted ABTS•+ solution was added to 100 mg of functionalized foil. After 15 and 60 min, the scavenging capability of ABTS•+ at 734 nm was calculated using Equation (3):Inhibition = (A_Control_ − A_Sample_)/A_Control_ · 100,(3)
where A_Control_ is the absorbance, measured at the starting concentration of ABTS•+, and A_Sample_ is the absorbance, of the remaining concentration of ABTS•+ in the presence of applicate chitosan–resveratrol onto the foils.

#### 2.4.3. Statistical Analysis

All the experimental data in this study are expressed as mean ± standard deviation and processed by Microsoft Excel 2016 (Microsoft Corporation, Redmond, Washington, DC, USA). One-way ANOVA analysis of variance was used in order to measure the statistical significance. Differences were taken to be significant for *p*  < 0.05. Statistically significant results are pointed out by *.

## 3. Results and Discussion

### 3.1. Dispersions Characterization

#### 3.1.1. Particle size, PDI, and Zeta Potential Determination

PDI and zeta potential of chitosan nanoparticles with embedded resveratrol were compared to those results of chitosan nanoparticles alone. Both types of dispersions had pH 4. The average size by intensity of synthesized nanoparticles determined with DLS measurements was 359 ± 40 nm for CSNPs and 950 ± 87 nm for CSNP’s RES, which is significantly different as seen from Table 2. This size increase clearly shows that RES was incorporated inside the particles and/or on the particles surface. PDI for CSNPs was 0.87 ± 0.09 and for CSNPs, RES was 0.58 ± 0.18, which suggests the higher monodispersity of CSNPs with incorporated RES, although there was no significant difference. Zeta potential measurements showed positive values and were 36 ± 5 mV for CSNPs and 48 ± 10 mV for CSNPs RES, respectively, with no significant difference. The raise in zeta potential values can be assigned to the presence of protonated amino groups at pH = 4 on the particles surface and also indicate the stability of nanoparticles dispersion [44]. The smaller increase in the positive ZP for nanoparticles dispersion with embedded resveratrol indicates that most of hydrophilic RES polyphenol was adsorbed into inner part of chitosan nanoparticles or onto chitosan nanoparticles surface in the way to provide sufficient steric or electrostatic repulsive forces among particles to ensure dispersion stability. 

Results of particle size and zeta potential of CSNPs and CSNPs RES are presented in Figure 1.

#### 3.1.2. MIC

MIC values of resveratrol (Table 3) were determined as an evaluation of their antimicrobial activity against selected bacteria. The method was used to test the ability of bacteria to produce visible growth when certain concentrations of resveratrol was added. The minimal inhibitory concentration of chitosan solutions and chitosan nanoparticles dispersion were determined; the results are listed in Table 4.

Obtained results are important in order to define the efficient concentration of resveratrol to be added during the synthesis of nanoparticles.

As it can be seen from Table 4, no significant differences are found in MIC for both bacteria; the gram-negative one, *Escherichia coli,* and the gram-positive one, *Staphylococcus aureus*.

### 3.2. Functional foils

#### 3.2.1. Gravimetric Measurements of Sample Mass

Differences in sample weight (**Δ*m***) using the gravimetric method evidently confirms the successful chitosan, chitosan nanoparticles-resveratrol application of the formulation onto the PE or PP foils (Table 5). In all cases, the mass of the samples increased, insignificantly, with respect to the reference foil. The effect of the hydrophobic foil and the application of hydrophilic colloidal systems as coatings showed poor adhesion and inhomogeneity. When the foil is treated with O_2_ plasma, the coating is much more homogenous. All these were clearly seen visually as well as by SEM as discussed later by SEM results. In all cases, foils were transparent after coatings.

#### 3.2.2. ATR-FTIR Spectroscopy

Typical peaks for PE and PP can be seen from the FTIR spectrum A1 and A2 (reference samples in Figure 2 and Figure 3). Details on FTIR spectrum of PE foil, labelled as A1, are shown in Figure 2. The wave numbers of FTIR spectra of PE at 2915, 2848, 1463, and 718.2 cm^−1^ were assigned to CH_2_ asymmetric stretching, CH_2_ symmetric stretching, bending deformation, and rocking deformation, respectively. Details of FTIR spectrum of PP foil (A2), are shown in Figure 2. The wave number of FTIR spectra of PP at 2950, 2917, 2867, 2842, 1458, 1376, 1167, 997, and 973 cm^−1^ were assigned to CH_3_ stretching, CH_2_ asymmetric stretching, CH_2_ symmetric stretching, CH_2_ symmetric stretching, CH_2_ bending vibration, CH_3_ bending vibration, CH_3_ symmetric deformation vibration, CH_3_ rocking vibration, and CH_2_ rocking vibration, respectively. The presence of mainly C–H vibrational groups clearly shows on the materials’ hydrophobicity.

Spectrum B1 (Figure 2 and Figure 3) represents the chitosan powder. A strong band in the region 3303–3347 cm^−1^ corresponds to N–H and O–H stretching, as well as the intramolecular hydrogen bonds. The absorption bands at around 2921 and 2877 cm^−1^ can be attributed to C–H symmetric and asymmetric stretching, respectively. The presence of residual *N*-acetyl groups was confirmed by the bands at around 1654 cm^−1^ (C=O stretching of amide I) and 1581 cm^−1^ (N–H bending of amide II); 1419 cm^−1^ belongs to the N–H stretching of the amide and the ether bonds and N–H stretching at 1375 cm^−1^ (amide III band), respectively. The CH_2_ bending and CH_3_ symmetrical deformations were confirmed by the presence of bands at around 1423 and 1375 cm^−1^, respectively. The absorption band at 1150 cm^−1^ can be attributed to asymmetric stretching of the C–O–C bridge. The bands at 1066 and 1026 cm^−1^ correspond to C–O stretching [45,46].

The FTIR spectrum of resveratrol showed a typical OH stretching band at 3182 cm^−1^ and intense bands at 1604 and 1583 cm^-1^ corresponding to C–C aromatic double bond stretching and C–C olefinic stretching, respectively. The peaks at 1513 and 1462 cm^−1^ reflect the benzene skeleton vibrations, while C–O stretching vibrations were seen by the peak at 1381 cm^−1^. The peaks at 986 and 964 cm^−1^ were ascribed to the bending vibration of C=C–H [47].

By comparing the peaks of reference foils (A1 and A2) with functionalized foils (D1, D2, E1, and E2), a successful application of coatings is seen. Spectrum D1 and D2 represent untreated PE and PP foils, which were firstly functionalized with solution of chitosan (2% w/v) and secondly with the chitosan–resveratrol nano dispersion (layer-by-layer). There are peaks on FTIR spectrum, which could confirm the presence of chitosan and resveratrol. The typical peaks of chitosan and resveratrol are smaller, but there are visible bonds such as N–H and O–H bond, all three amide bonds, C=C alkene, C=C aromatic, C–O phenols, and =C–H aromatic bonds. It must be pointed out that the coating on the films (spectra D1 and D2) were not homogeneous, due to the hydrophobic character of foils and the hydrophilic nature of chitosan and chitosan–resveratrol formulation. Due to the difference in polarity, the adhesion of applied formulations is lowered and not uniform.

All these peaks can also be observed on the spectra E1 and E2, but with the significant higher intensity. For all the coated PE and PP films, the characteristic peaks of chitosan were observed (N–H and O–H stretching; amide I, amide II, and amide III; C–O stretching). Comparison of the FTIR spectra between D1 and E1 (by normalizing) is shown in the Figure 2b. The bands corresponding to C–O stretching (glycoside bonds) and C=O stretching (amide I) are shifted towards higher wavenumber (1096 and 1675 cm^−1^) implying the linkage of CSNPs and CSPSs RES with the oxygen-rich foil with respect to unmodified foil (plasma-non activated). In addition, the vibration frequencies of N–H and O–H bands are also shifted towards higher wavenumber (3310 and 3383 cm^−1^) and the peak is more pronounced with larger intensity for O_2_ modified foil with CSNPs RES applied formulation. The increased band vibrations frequency of N–H and O–H bonds suggest that the interactions between the chitosan groups and foils surface are present in larger amount. It might be concluded that with O_2_ plasma treatment the new binding sites (the appearance of oxygen-rich groups on PE surface such as CO, COOH, OH) are yielded onto foils surface, which are available for chemically binding with amino groups of chitosan and, consequently, more chitosan–resveratrol nano dispersion could be attached on this first bounded layer during a second coating step. When FTIR spectra between D2 and E2 for PP foils are compared (Figure 3b), a similar trend was obtained, i.e., the occurrence of oxygen containing groups with O_2_ plasma treatment led to better adsorption of chitosan as well as further attachment of chitosan nanoparticles with embedded resveratrol.

#### 3.2.3. XPS Analysis

The surface chemical composition of the reference foils and plasma-functionalized foils determined by XPS is presented in Table 6. For the untreated PE and PP foils (A1 and A2), mostly carbon was detected, which is in the agreement with chemical composition of the virgin foils. Small amounts of oxygen (approximately 1 at.%) are due to the surface contamination as often observed in XPS spectra measured on pure polymer foils. After oxygen plasma treatment, the oxygen concentration significantly increased to 13.7 ± 0.3 and 14.77 ± 0.1 at.% for PE and PP, respectively (samples A3 and A4 in Table 6). Slightly higher oxygen concentration on PP sample can be explained by oxidation of the side methyl group, which is absent on PE. Increased oxygen concentration clearly indicates successful surface plasma activation of both polymer foils with introduction of oxygen functional groups that may act as a binding place for chitosan. This is further proved by Figure 4 showing formation of hydroxyl, carbonyl, and carboxyl groups on the surface of the PE and PP foils.

When untreated (plasma none activated) foils were coated with chitosan and chitosan–resveratrol nano dispersion, nitrogen appears on the surface of the foils; i.e., 3.3 at.% for PE (D1) and 2.1 at.% for PP (D2). According to chemical structure, only chitosan contains nitrogen whereas resveratrol should not possess any nitrogen; therefore, nitrogen is explained by the presence of amino groups originating from chitosan because of its attachment on the foils. It can also be observed, that the carbon content is reduced, whereas the oxygen content is increased by 8% for PE (D1) and 7.4% for PP (D2) in comparison to the untreated foils, which is explained by a high oxygen content in both organic substances i.e., chitosan and resveratrol.

Functionalized foils that were previously treated with oxygen plasma show the highest nitrogen content, indicating a higher amount of adsorbed chitosan on the surface as compared to the untreated foils. This may be due to the chemisorption of prepared chitosan dispersion onto foils. As it has been shown, oxygen plasma introduced carbonyl and carboxyl group that may form binding places for chitosan amino groups [16]. Oxygen functional groups formed by plasma treatment also raised hydrophilicity of the foils causing better wetting of the surface with chitosan solution. All these may improve the ability for chitosan adsorption, i.e., higher amount of chitosan could be bounded, which was clearly indicated in our case by a higher concentration of nitrogen; i.e., nitrogen concentration significantly increased for 6.7% on PE surface (E1) and for 6.7% on PP surface (E2). The carbon content considerably decreased by as much as 25.7% on PE (E1) and 25.8% on PP (E2), and the oxygen content significantly increased by as much as 18% on PE (E1) and PP (E2) indicating attachment of chitosan and resveratrol, which are both rich with oxygen. (E2). All these observations are in a good agreement with the results, obtained from the infrared spectroscopy.

Summarizing, when all plasma-functionalized samples are compared to those that were previously not activated by plasma via reference ones, it is clearly observed that the amount of nitrogen, originating from the coating, is significantly increased, which means that chitosan has adsorbed onto plasma-treated foil to a much higher extent.

#### 3.2.4. Oxygen Permeability

Table 7 contains the results of oxygen permeability for the reference foils compared to the functionalized foils. Oxygen permeability after coatings of foils has been reduced in all cases. The permeability decrease for the plasma untreated PE and PP foils in comparison with reference samples was lower than for the same samples, which were previously activated by O_2_ plasma. This may be attributed to the higher and uniform amount of attached chitosan macromolecular solution as the first layer and, thus, further better affinity for chitosan-RES nanoparticles as a second layer. The statistical analysis showed that the highest significant decrease in permeability was observed for O_2_ plasma-treated PE foil with a layer of chitosan and a layer of chitosan–resveratrol nano dispersion (E1). For this sample, the oxygen permeability was reduced from 3226 ± 62 cm^3^/m^2^d by as much as 202 ± 16 cm^3^/m^2^d. Significantly different results were also obtained for the plasma-functionalized PP foil (E2), as oxygen permeability decreased from 1078 ± 36 cm^3^/m^2^d to 195 ± 14 cm^3^/m^2^d. For plasma-untreated PE and PP foils coated with a layer of chitosan and a second layer of chitosan–resveratrol nano dispersion, the maximum permeability of oxygen has non-significantly reduced from 3226 ± 62 cm^3^/m^2^d to 2417 ± 104 cm^3^/m^2^d for sample D1 and from 1078 ± 36 cm^3^/m^2^d to 968 ± 19 cm^3^/m^2^d for sample D2. The latest may be attributed to the lower, and especially homogenous, adsorption of chitosan colloidal systems onto both plasma non-activated foils, as will be discussed in Section 3.2.6.

#### 3.2.5. Goniometry

Static contact angle (SCA) measurements, which provides an inverse measure of wettability, have also been performed. SCA of various samples are also shown in Figure 5. On the basis of SCA results, hydrophilicity/hydrophobicity of PE and PP surfaces was determined. Reduction of contact angle is of a great importance for practical use as the hydrophilic surface of the foils reduces the potential process of dew condensation on the surface of the foils (anti-fog efficiency) in contact with food, which worsens the packaging conditions and thus increases the food contamination [48]. It is, thus, extremely important to avoid this process.

The results of SCA measurements are presented in Table 8. The reference PE and PP foils had contact angles of 108.2° and 109.3°, which pointed out foils hydrophobicity. Statistically, all samples are significantly different compared to A1 or A2 (control sample). The treatment with oxygen plasma was successful as expected and resulted into contact angle decrease for around three to four times (A3, A4). With the functionalization of the foils using chitosan systems, the SCA significantly decreased in comparison to reference foils (A1 and A2). The contact angles slightly decreased for the samples D1, D2; however, the biggest significant difference in the decrease of the contact angle was observed for the samples E1 and E2, i.e., contact angle decreased down to 42.4° and 45.3°. It can be concluded that when foils (PE or PP) are previously treated with O_2_ plasma, the higher and more homogeneous adsorption of chitosan macromolecular solution and chitosan–resveratrol nano dispersion occurred. Because of the adsorbate foils polarity and hydrophilicity, it is obvious that coatings on the plasma-activated PE and PP surface has better wettability and are thus more homogeneous after attachment.

#### 3.2.6. Morphology: Scanning Electron Microscopy—SEM

Figure 6 presents SEM images of the surface morphology and the adhesion uniformity of the uncoated (reference samples A1 and A2) and coated foils with chitosan in the first layer and CSNPs RES formulation in the second (upper) layer. For easier comparison, all images were taken at the comparable magnification (300×) and at the same imaging parameters. For samples D1 and D2, the coating is visible on the surface of the foils with significantly higher contrast together with the deposited agglomerates. However, the coating was rough and did not completely cover the foil as it can be seen in comparison to the reference foils. This can be explained by the hydrophobicity of the two polymers (the hydrophobicity of the reference PE and PP foils were demonstrated by measuring the contact angle as well as surface techniques that suggested the presence of nonpolar groups) that lowers the adhesion of applied formulations due to the difference in polarity.

With O_2_ plasma treatment/foils surface modification before coating, the surface of the PE and PP foils were modified. The latter significantly improved the interaction between the two polymers (PE/PP foil and chitosan). When the reference foils are compared with the samples E1 and E2 (Figure 6), a very successful application is obtained as the uniform hydrophilic surface, where the coating is homogenous, smoother, and thinner, uniformly covering the foil with less observed agglomerates. It was found by surface analyses (i.e., infrared spectroscopy and XPS) that the occurrence of oxygen-containing functional groups (C═O, C–O, and –OH) of the O_2_ plasma-treated PE and PP foils increased from those of the untreated one, indicating that the O_2_ plasma enhanced hydrophilicity of the PE and PP foils (E1 and E2). The SEM analysis also supported the oxygen permeability results, where the homogeneous formulation coating (samples E1 and E2) also showed higher reduced permeability with respect to samples D1 and D2. This can be in part explained by the more homogenous adhesion of the applied layer-by-layer coating, as evident from SEM images.

#### 3.2.7. Migration: Desorption Experiment—Polyelectrolyte Titration

Chitosan may be released from the PE and PP surface if foils are treated in a way that the interactions between both polymers are weakening. Desorption experiments were performed at pH = 5.8 (mimicking the pH of most of the food). Prior to the polyelectrolyte titration, pH of bath solution was adjusted by 0.1 M HCl to 3.6. Polyelectrolyte titration results shows how much chitosan desorbed from the surface of PE and PP after 24 h. Table 9 shows the amount of amino groups of chitosan that were desorbed from one gram of PE or PP foil in the indicated range of five repetitions with standard deviation within interval of 0.5%–3%. If the previously untreated foils (D1 and D2) are compared with the treated ones with oxygen plasma (E1 and E2), it can be seen that less amino groups are desorbed from the latter ones.

Namely, for samples D1 and D2, 0.0189 mmol and 0.0118 mmol of amino groups per gram of foil were desorbed, respectively, whilst for E1 and E2, 0.0019 mmol and 0.0051 mmol amino groups per gram of foil. This, again, suggests better adhesion of coating formulations on previously modified foils with O_2_ plasma with stronger interactions among bonding sites (ionic interactions). The amount of desorbed chitosan in mmol/g is shown in Table 9 and the example of polyelectrolyte titration in Figure 7.

#### 3.2.8. Bioactivity

##### Antimicrobial Activity

Using antimicrobial testing, the chitosan/chitosan–resveratrol nano dispersion-coated PE and PP foils were tested against gram-positive (*Staphylococcus aureus)* and gram-negative bacteria (*Escherichia coli*). The results of antimicrobial efficacy are presented in Table 10.

It is known that the protonated amino groups from chitosan structure play a crucial role in the antibacterial activity. It has to be pointed out that there are two layers of coatings on the foils, containing 2% chitosan and chitosan–resveratrol nano dispersion. For resveratrol, which is a stilbenoid, a type of natural phenol, and a phytoalexin, significant and strong antioxidant properties are known with simultaneous antimicrobial efficiency [33].

Results in Table 10 show that by functionalized foils surface, antimicrobial efficiency is achieved in all cases. As already mentioned, the application on the samples D1 and D2 is non-homogeneous; therefore, the values of reduction are different. Somewhere they are lower (D2), but in the case of D1, high antimicrobial efficacy is seen. Very high and statistically significant antimicrobial activity is visible for the foils previously treated with O_2_ plasma. It can be seen for both samples E1 and E2, that antimicrobial efficacy for bacteria *Staphylococcus aureus* is more than 90%. The same functionalized foils also have high and statistically significant antimicrobial efficiency against bacteria *Escherichia coli*; samples E1 and E2, showed reduction of 77.55 ± 2.57% and 79.43 ± 5.86%, respectively.

##### Anti-Oxidant activity—ABTS

Figure 8 shows the results of anti-oxidative activity. The test was used to confirm how functionalized packaging influence the oxidative process inhibition. As can be seen from the results presented in Figure 8, the lowest anti-oxidative activity is shown for the reference PE (4.00%) and PP (5.44%) foils. Each applied foil has a greatly increased effect, which means that resveratrol is a very good antioxidant. Good anti-oxidative activity results mean that resveratrol has been successfully loaded to chitosan nano dispersion, whilst chitosan itself does not possess antioxidant properties [30].

The results for samples D1 (89.68%), D2 (95.72%), E1 (100%), and E2 (100%) after 15 min, show very good inhibition on all applied foils. Because they were applied onto the foils in layers, which means that after the 2% chitosan adsorption a layer of chitosan–resveratrol nano dispersion was applied onto dry foil. This is, therefore, the external layer of the foil surface and as such, easily accessible for free radicals. In cases E1 and E2, a 100% inhibition is obtained after 15 min; in other cases, (D1 and D2) after 60 min.

## 4. Conclusions

It was demonstrated that the additive effect of antimicrobial chitosan together with anti-oxidative resveratrol as an adsorbate for PP and PE, introduces bioactive properties to foils surface. XPS and FTIR methods show successful binding of all chitosan and chitosan–resveratrol structured adsorbates onto foils. This is also supported by the differences in sample weight after coating application onto foils using gravimetric method. Both attached layers (chitosan macromolecular solution and further dispersion of chitosan nanoparticles with embedded resveratrol) lowered foils oxygen permeability as well as reduced contact angle, which indicates great barrier and good anti-fog foils properties.

With O_2_ plasma treatment, the surface of the PE and PP foils were modified, and significant improvement of the interaction between the two polymers, PE/PP foil and chitosan, was achieved with very successful coating application. It was found by surface analyses (FTIR and XPS) that the occurrence of oxygen-containing functional groups (C═O, C–O, and −OH) of the O_2_ plasma-treated PE and PP foils increased from those of the untreated one, indicating that the O_2_ plasma-enhanced hydrophilicity of the PE and PP foils resulting into more homogeneous formulation coating as evident from SEM images.

The coatings were smoother and thinner, uniformly covering the foils with less observed agglomerates as in the case of plasma non-activated foils. Moreover, plasma-activated and further coated foils by two mentioned layers show much higher reduction of oxygen permeability as well as much better anti-fog efficiency due to lower contact angle. Besides all mentioned and good applicable properties, these functionalized foils also have superior antimicrobial and anti-oxidative properties. The method of previous plasma activation of foils and further functionalization by chitosan macromolecular solution as a first layer and chitosan nanoparticles with integrated resveratrol as a second layer, shows great potential for different packaging applications such as active packaging in food industry.

## Figures and Tables

**Figure 1 materials-12-02118-f001:**
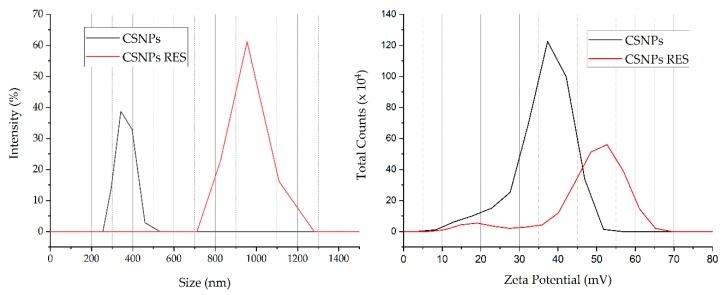
Particle size, PDI, and zeta potential of CSNPs and CSNPs RES.

**Figure 2 materials-12-02118-f002:**
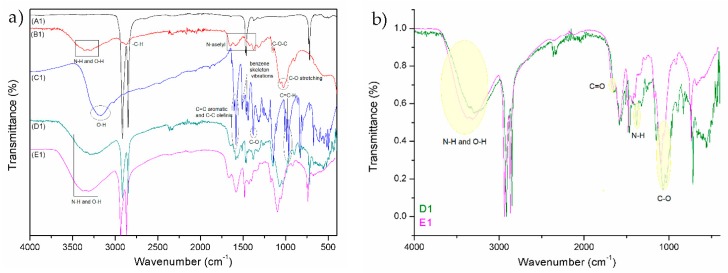
FTIR spectra of reference polyethylene (PE) (A1), chitosan (B1), resveratrol (C1), and different treated PE foils with application (D1 and E1) (**a**); comparison of the FTIR spectra between D1 and E1 (by normalizing) (**b**).

**Figure 3 materials-12-02118-f003:**
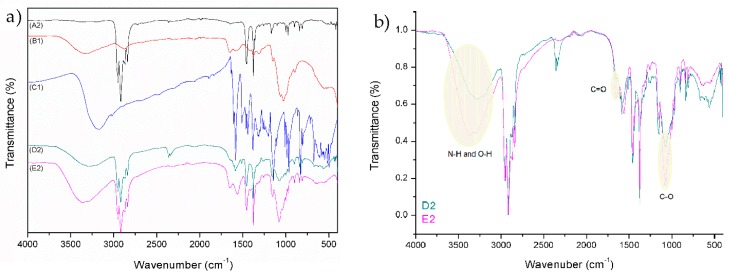
FTIR spectra of reference polypropylene (PP) (A2), chitosan (B1), resveratrol (C1), and different treated PP foils with application (D2 and E2) (**a**); comparison of the FTIR spectra between D2 and E2 (by normalizing) (**b**).

**Figure 4 materials-12-02118-f004:**
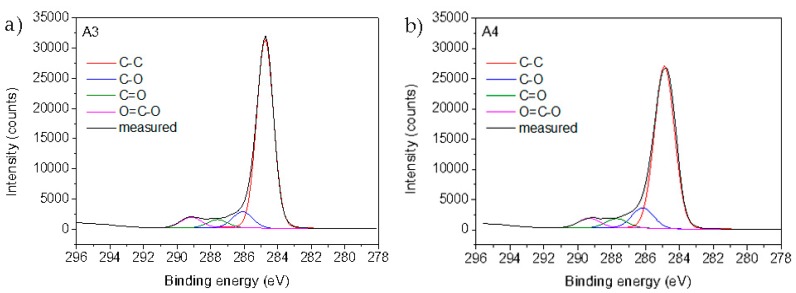
High-resolution carbon C1s spectrum of plasma-treated (**a**) PE and (**b**) PP samples.

**Figure 5 materials-12-02118-f005:**
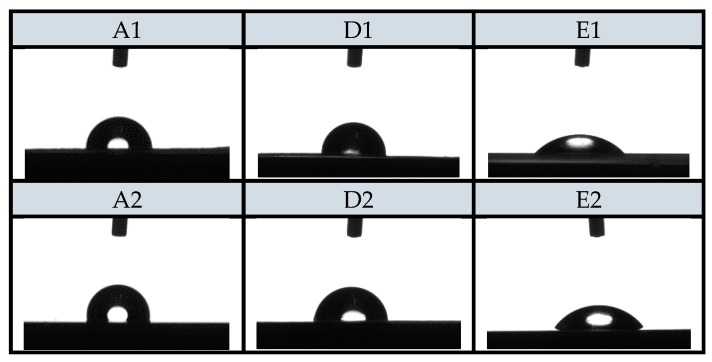
CSA of samples.

**Figure 6 materials-12-02118-f006:**
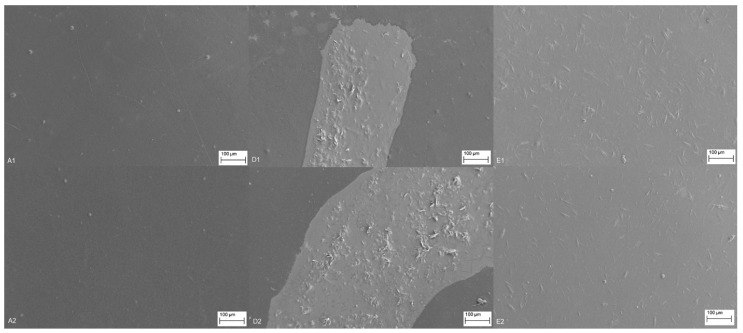
SEM micrographs of functionalized surface of PE and PP with resveratrol-loaded chitosan- sodium tripolyphosphate (TPP) nano dispersion.

**Figure 7 materials-12-02118-f007:**
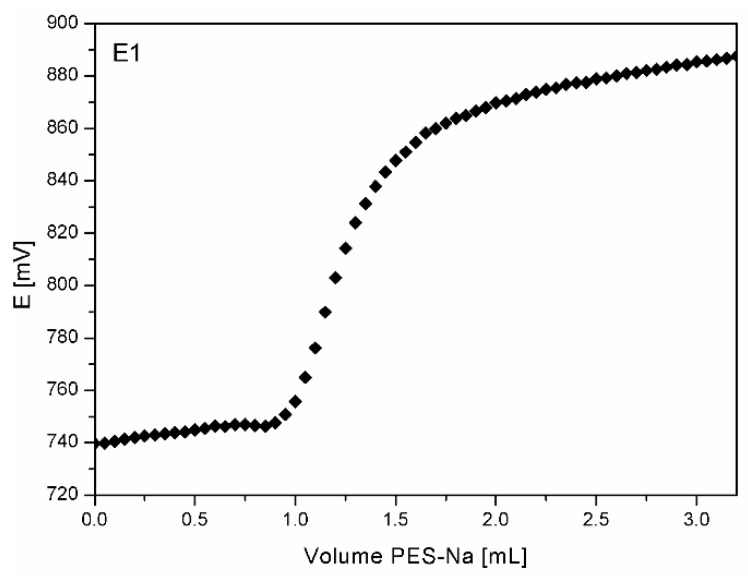
Polyelectrolyte titration curve on the example of sample E1 after 24 h.

**Figure 8 materials-12-02118-f008:**
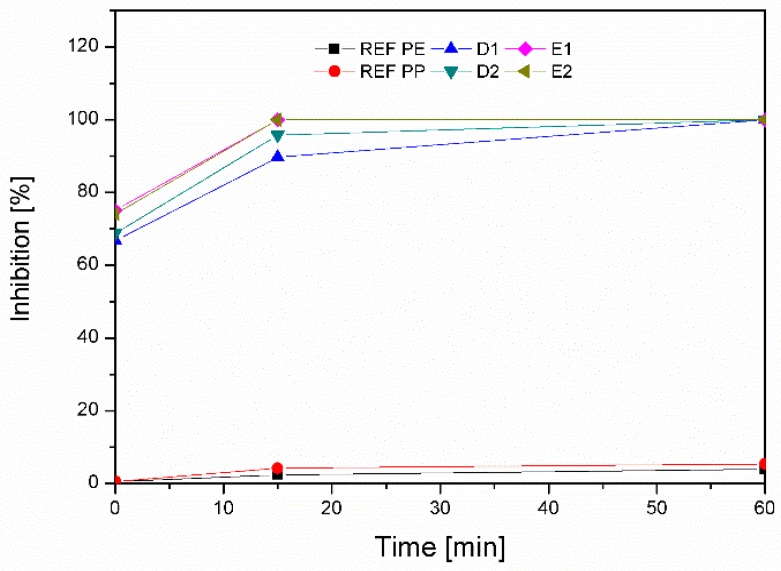
Graphical representation of anti-oxidative activity of CS, CSNPs RES-coated PE, and PP foils after the same time interval.

**Table 1 materials-12-02118-t001:** Samples descriptions.

Sample Notation	Description of Sample
A1	polyethylene (PE) foil
A2	polypropylene (PP) foil
A3	PE foil treated with O_2_ plasma
A4	PP foil treated with O_2_ plasma
B1	chitosan powder (CS)
B2	2% (w/v) chitosan (CS)
B3	chitosan nanoparticles (CSNPs) dispersion
C1	resveratrol powder (RES)
C2	Chitosan–resveratrol nano dispersion (CSNPs RES)
D1	untreated PE foil, applicate with CS-1. layer and CSNPs RES-2. layer
D2	untreated PP foil, applicate with CS-1. layer and CSNPs RES-2. layer
E1	PE foil treated with O_2_ plasma, applicate with CS-1. layer and CSNPs RES-2. layer
E2	PP foil treated with O_2_ plasma, applicate with CS-1. layer and CSNPs RES-2. layer

**Table 2 materials-12-02118-t002:** Particle size, polydispersity index (PDI) and zeta potential of chitosan nanoparticles (CSNPs) and CSNPs resveratrol (RES).

Sample	Z-Average (nm) by Intensity	PDI	ZP (mV)	pH
CSNPs	359 ± 40	0.87 ± 0.09	36 ± 5	4.0
CSNPs RES	**950 ± 87 ***	0.58 ± 0.18	48 ± 10	4.0

**Table 3 materials-12-02118-t003:** Minimal inhibitory concentration (MIC) (mg/mL) of resveratrol determined by microdilution method.

Bacteria	MIC (mg/mL)
*Escherichia coli*	5.01
*Staphylococcus aureus*	0.16

**Table 4 materials-12-02118-t004:** Minimal inhibitory concentrations of chitosan solutions (CS) and dispersion of chitosan nanoparticles (CSNPs).

Bacteria	MIC (mg/mL) of CS	MIC (mg/mL) of CSNPs
*Escherichia coli*	0.0053 ± 0.0011	0.0092 ± 0.0029
*Staphylococcus aureus*	0.0039 ± 0.0001	0.0078 ± 0.0002

**Table 5 materials-12-02118-t005:** Percentage (%) of adsorption of chitosan macromolecular solution and CSNPs RES dispersion onto the foil.

Sample	Mass of Dry Reference Foil [g/cm^2^]	Mass of Dry Functionalised Sample [g/cm^2^]	Δ*m* [g/cm^2^]	Adsorption [%]
D1	46.36 ± 0.03	46.70 ± 0.01	0.34 ± 0.02	0.73
D2	23.20 ± 0.06	23.52 ± 0.08	0.32 ± 0.07	1.36
E1	46.32 ± 0.02	46.49 ± 0.02	0.17 ± 0.02	0.36
E2	23.23 ± 0.01	23.50 ± 0.04	0.27 ± 0.02	1.18

**Table 6 materials-12-02118-t006:** Surface chemical composition (at.%) of the reference foils and plasma-treated foils with the two-layer coating (at.%). Statistical significance is defined as * *p* < 0.05 compared to control sample (ANOVA test).

Untreated Foils				O_2_ Treated Foils			
Sample	C	N	O	Sample	C	N	O
A1	98.9 ± 0.6	-	1.1 ± 0.3	A3	**86.3 ± 0.3 ***	-	**13.7 ± 0.3 ***
A2	98.7 ± 0.3	-	1.3 ± 0.1	A4	**85.3 ± 0.1 ***	-	**14.7 ± 0.1 ***
D1	**87.7 ± 2.6 ***	3.3 ± 0.8	**9.1 ± 2.0 ***	E1	60.6 ± 0.0	6.7 ± 0.2	31.7 ± 0.3
D2	**89.7 ± 0.8 ***	2.0 ± 0.1	**8.3 ± 0.8 ***	E2	59.5 ± 0.1	6.7 ± 0.0	32.7 ± 0.5

**Table 7 materials-12-02118-t007:** The oxygen permeability between the reference foils and the foils with the application. Statistical significance is defined as * *p* < 0.05 compared to control sample (ANOVA test).

Sample	OTR (cm^3^/m^2^d)
A1	3226 ± 62
A2	1078 ± 36
D1	2417 ± 104
D2	968 ± 19
E1	**202 ± 16 ***
E2	**195 ± 14 ***

**Table 8 materials-12-02118-t008:** Values of static contact angle (SCA) measurements. Statistical significance is defined as * *p* < 0.05 compared to control sample (ANOVA) test).

Sample	Average Angle (α/°)	Difference (%)
A1	108.2 ± 1.2	/
A2	109.3 ± 0.7	/
A3	**29.70 ± 0.5 ***	72.6
A4	**37.0 ± 0.5 ***	66.1
D1	**93.1 ± 1.2 ***	14.0
D2	**84.8 ± 1.4 ***	22.4
E1	**42.4 ± 1.8 ***	60.8
E2	**45.3 ± 0.9 ***	58.6

**Table 9 materials-12-02118-t009:** Desorption of chitosan amino group per gram of foil.

Desorption of Chitosan	Sample			
	D1	D2	E1	E2
[mmol/kg]	18.90 ± 2.84	11.80 ± 1.30	1.92 ± 0.42	0.51 ± 0.31

**Table 10 materials-12-02118-t010:** Antimicrobial efficacy of gram-positive (*Staphylococcus aureus*) and gram-negative bacteria (*Escherichia coli*). Statistical significance is defined as * *p* < 0.05 compared to control sample (ANOVA test).

Bacteria	*Staphylococcus aureus*		*Escherichia coli*	
Sample	Number of Cells	Antimicrobial Efficacy	Number of Cells	Antimicrobial Efficacy
	(log cfu/cm²)	(%)	(log cfu/cm²)	(%)
A1	4.58 ± 0.06	-	4.58 ± 0.06	-
A2	5.10 ± 0.05	-	5.10 ± 0.05	-
D1	1.35 ± 0.30	72.91 ± 8.26	3.25 ± 0.14	37.73 ± 2.77
D2	3.57 ± 0.33	22.03 ± 7.28	3.89 ± 0.36	25.32 ± 6.87
E1	**0.67 ± 0.21 ***	93.54 ± 3.95	**1.24 ± 0.09 ***	77.55 ± 2.57
E2	**0.51 ± 0.08 ***	91.21 ± 2.22	**1.07 ± 0.31 ***	79.43 ± 5.86

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
