# Peer review of "Functionalization of Polyethylene (PE) and Polypropylene (PP) Material Using Chitosan Nanoparticles with Incorporated Resveratrol as Potential Active Packaging"

_materials, 2019, doi:10.3390/ma12132118_

Round 1
Reviewer 1 Report
Please include images for DLS and zeta potential figures.
Please include clear SEM images
If possible, please include images of antimicrobial assay figures
Please rewrite conclusion and make it as a single paragraph.
Please include some recent references from last two years.
Author Response
Dear reviewer,
Please find below our response to the comments of reviewer regarding our manuscript. We would like to thank the reviewers for their valuable comments, which helped us to improve the manuscript. All cited issues have been clarified to the best of our knowledge. We hope that the article is now suitable for publication. The modifications to the original manuscript are tracked through changes in the revised version. In addition, please find our answers to the comments below.
We believe that these modifications improved the manuscript significantly and we hope that they will meet your expectations.
Sincerely yours, Prof. Dr. Lidija Fras Zemljič (as corresponding author).
...............................................................................................................................
RESPONSE TO REVIEWER 1:
Comments and Suggestions for Authors
Please include images for DLS and zeta potential figures.
Answer: Thank you for your suggestions. The figures were included.
Please include clear SEM images
Answer: Thank you for the constructive suggestion. At our University, we have access only to SEM where all optimised parameters were used for polymer materials, thus repetition of experiment would give similar micrographs. SEM was used to analyse and compare morphology and coating adhesion of foils to the modified ones. The deposition of the chitosan nanoparticles alone or with embedded polyphenols was clearly successful as can be seen from the SEM micrographs. However, for samples with non-previous activation of plasma, the coating is visible on the surface of the foils with significantly higher contrast together with the deposited agglomerates. The coating was rough and did not completely cover the foil as it can be seen in comparison to the reference foils. This can be explained by the hydrophobicity of the two polymers (the hydrophobicity of the reference PE and PP foils were demonstrated by measuring the contact angle as well as surface techniques which suggested the presence of nonpolar groups) that lowers the adhesion of applied formulations due to the difference in polarity. With O2 plasma treatment/foils surface modification before coating, the surface of the PE and PP foils were modified, and the latter significantly improved the interaction between the two polymers (PE/PP foil and chitosan); a very successful application is obtained as the uniform hydrophilic surface, where the coating is homogenous, smoother and thinner, uniformly covering the foil with less observed agglomerates.
Note: The non- or barely conductive properties of materials causing overcharging and subsequently contribute to the “blurring” of the images is in general a very common issue, which unfortunately persisted when obtaining scans.
If possible, please include images of antimicrobial assay figure
Answer: Thank you for suggestion. Unfortunately, we do not possess those pictures.
Please rewrite conclusion and make it as a single paragraph.
Answer: Thank you for suggestion. The conclusion was rewritten as suggested.
Please include some recent references from last two years.
Answer: Thank you for suggestion. The references were updated.
Reviewer 2 Report
This manuscript describes a novel method for improving the properties of polyethylene (PE) and polypropylene (PP) polymer foils suitable for applications in a food packaging. XPS and FTIR methods show binding of all chitosan and chitosan-resveratrol structured adsorbates onto foils, and it was demonstrated that the additive effect of antimicrobial chitosan together with anti-oxidative resveratrol, as an adsorbate for PP and PE foils with previous oxygen plasma activation may be more accompanied with more homogeneous structure when applied onto their surfaces. However, the experiments are lack of novelty, and the results cannot support the conclusions very well. Publication is not recommended.
Author Response
Dear reviewer,
Please find below our response to the comments of reviewer regarding our manuscript. We would like to thank the reviewers for their valuable comments, which helped us to improve the manuscript. All cited issues have been clarified to the best of our knowledge. We hope that the article is now suitable for publication. The modifications to the original manuscript are tracked through changes in the revised version. In addition, please find our answers to the comments below.
We believe that these modifications improved the manuscript significantly and we hope that they will meet your expectations.
Sincerely yours, Prof. Dr. Lidija Fras Zemljič (as corresponding author).
---------------------------------------------------------------------
RESPONSE TO REVIEWER 2:
Comments and Suggestions for Authors
This manuscript describes a novel method for improving the properties of polyethylene (PE) and polypropylene (PP) polymer foils suitable for applications in a food packaging. XPS and FTIR methods show binding of all chitosan and chitosan-resveratrol structured adsorbates onto foils, and it was demonstrated that the additive effect of antimicrobial chitosan together with anti-oxidative resveratrol, as an adsorbate for PP and PE foils with previous oxygen plasma activation may be more accompanied with more homogeneous structure when applied onto their surfaces. However, the experiments are lack of novelty, and the results cannot support the conclusions very well. Publication is not recommended.
Answer: Thank you for comments. The article was a bit modified and we believe that description of a novel method is presented in a way to show novelty and it is interesting enough for readers of the Journal. This (novelty) can also be supported by the fact that our research work regarding this topic is patent application.
Reviewer 3 Report
The manuscript deals with the functionalization of polyethylene (PE) and polypropylene (PP) material using chitosan nanoparticles with incorporated resveratrol as potential active packaging.
The English language must be revised.
Please separate values from units, e.g. “25 ºC” not “25ºC”.
Abstract
This section is vague. Please add your main results.
Materials and methods
A statistical analysis section is missing.
Water vapor permeability???
Color???Opacity???
Results and discussion
Please add different superscript letters for significant differences in all results and revise the discussion in accordance.
Conclusion
This section is short and must be improved.
Author Response
Dear reviewer,
Please find below our response to the comments of reviewers regarding our manuscript. We would like to thank the reviewers for their valuable comments, which helped us to improve the manuscript. All cited issues have been clarified to the best of our knowledge. We hope that the article is now suitable for publication. The modifications to the original manuscript are tracked through changes in the revised version. In addition, please find our answers to the comments below.
We believe that these modifications improved the manuscript significantly and we hope that they will meet your expectations.
Sincerely yours, Prof. Dr. Lidija Fras Zemljič (as corresponding author).
---------------------------------------------------------------------------------
RESPONSE TO REVIEWER 4:
The manuscript deals with the functionalization of polyethylene (PE) and polypropylene (PP) material using chitosan nanoparticles with incorporated resveratrol as potential active packaging.
The English language must be revised.
Answer: Thank you for comments. The English language was revised from the native speaker.
Please separate values from units, e.g. “25 ºC” not “25ºC”.
Answer: Thank you for comments. Values were separated from units.
Abstract: This section is vague. Please add your main results.
Answer: Thank you for comments. The abstract was corrected as suggested.
Materials and methods
A statistical analysis section is missing.
Answer: Thank you for comments. The data were inserted where possible.
Water vapor permeability???
Color???Opacity???
Answer: Thank you for comments. Water vapour permeability and colour opacity were not researched in this article. The foils were transparent after modification and this was added into results section. The main parameter as anti-fog condensation capacity were discussed through contact angle determination and barrier properties through determination of oxygen permeability. The oxygen is the most critical for food spoilage thus to our opinion oxygen permeability - parameter is enough to be presented.
Results and discussion
Please add different superscript letters for significant differences in all results and revise the discussion in accordance.
Answer: Thank you for comments. To our opinion, the results and discussion are well structured and understandable for readers. The results of each techniques and suitable properties determined were pointed out divided into subchapters. In each subchapter, the most important facts and significant differences among samples were pointed out.
Conclusion
This section is short and must be improved.
Answer: Thank you for comments. This part was corrected as suggested.
Round 2
Reviewer 2 Report
It can be accepted in present form
Author Response
Thank you for your approval. The article was upgraded through 2 revisions so it is now very much improved.
Reviewer 3 Report
Introduction
“However, in spite of chitosan excellent antimicrobial activity, it has been found that chitosan shows very poor antioxidant activity [22,23], required for a bioactive packaging material in order to impede the natural processes leading to food spoilage by reducing oxygen and moisture.”??? Please rephrase.
Materials and methods
A statistical analysis section is still missing.
Results and discussion
Please add different superscript letters for significant differences in all results and revise the discussion in accordance.
Author Response
All the answers are given in attachment. The changes are tracked green.
Thank you.

Round 3
Reviewer 3 Report
The manuscript was improved.